# iSparse: Output Informed Sparsification of Neural Networks

## Abstract

Deep neural networks have demonstrated unprecedented success in various knowledge management applications. However, the networks created are often very complex, with large numbers of trainable edges which require extensive computational resources. We note that many successful networks nevertheless often contain large numbers of redundant edges. Moreover, many of these edges may have negligible contributions towards the overall network performance. In this paper, we propose a novel *iSparse framework*, and experimentally show, that we can sparsify the network, by 30-50%, without impacting the network performance. iSparse leverages a novel edge significance score, $E$, to determine the importance of an edge with respect to the final network output. Furthermore, iSparse can be applied both while training a model or on top of a pre-trained model, making it a retraining-free approach - leading to a minimal computational overhead. Comparisons of iSparse against PFEC, NISP, DropConnect, and Retraining-Free on benchmark datasets show that iSparse leads to effective network sparsifications.

## 1 Introduction

Deep neural networks (DNNs), particularly convolutional neural networks (CNN), have shown impressive success in many applications, such as facial recognition (Lawrence et al., 1997), time series analysis (Yang et al., 2015), speech recognition (Hinton et al., 2012), object classification (Liang & Hu, 2015), and video surveillance (Karpathy & et. at., 2014). As the term *"deep" neural networks implies*, this success often relies on large networks, with large number of trainable edges (weights) (Huang et al., 2017; Zoph et al., 2018; He et al., 2016; Simonyan & Zisserman, 2015).

While a large number of trainable edges help generalize the network for complex and diverse patterns in large-scale datasets, this often comes with enormous computation cost to account for the non-linearity of the deep networks (ReLU, sigmoid, tanh). In fact, DNNs owe their recent success to hardware level innovations that render the immense computational requirements practical (Ovtcharov & et. al., 2015; Matthieu Courbariaux et al., 2015). However, the benefits of hardware solutions and optimizations that can be applied to a general purpose DNN or CNN are limited and these solutions are fast reaching their limits. This has lead to significant interest in network-specific optimization techniques, such as network compression (Choi & et. al., 2018), pruning (Li et al., 2016; Yu et al., 2018), and regularization (Srivastava & et. al., 2014; Wan et al., 2013), aim to reduce the number of edges in the network. However, many of these techniques require retraining the pruned network, leading to the significant amount of computational waste.

### 1.1 Network Sparsification

Many successful networks nevertheless often contain large numbers of redundant edges. Consider for example, the weights of sample network shown in Figure 1a. As we see here, the weight distribution is centered around zero and has significant number of weights with insignificant contribution to the network output. Such edges may add noise or non-informative information leading to reduction in the network performance. (Denil et al., 2013; Ashouri et al., 2018; Yu et al., 2018) has shown that it is possible to predict 95% network parameters while only learning 5% parameters.

Sparsification techniques can generally be classified into neuron/kernel sparcification (Li et al., 2016; Yu et al., 2018) and edge/weight sparcification techniques (Wan et al., 2013; Ashouri et al.,

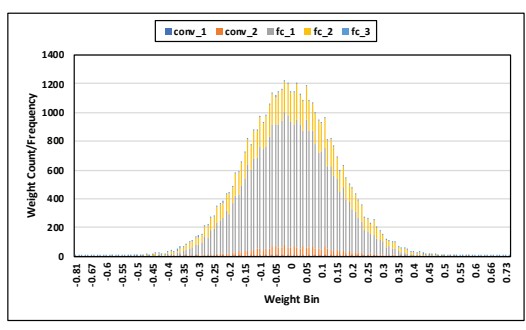 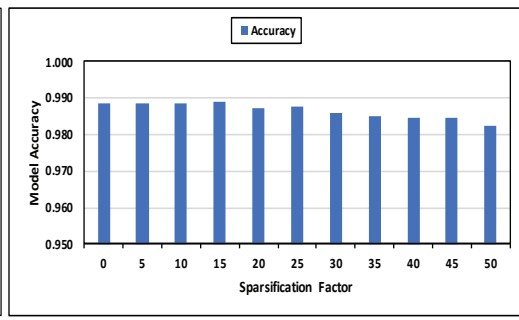

(a) Weight Distribution for LeNet-5          (b) Model Accuracy vs Sparsification Factor

Figure 1: Overview of weight distribution and model accuracies for MNIST dataset (LeNet-5)

2018): (Li et al., 2016) proposed to eliminate neurons that have low *l2*-norms of their weights, whereas (Yu et al., 2018) proposed a neuron importance score propagation (NISP) technique where neuron importance scores (using Roffo & et. al. (2015) - See Equation 5) are propagated from the output layer to the input layer in a back-propagation fashion. Drop-out (Srivastava & et. al., 2014) technique instead deactivates neuron activations at random. As an edge sparsification technique, DropConnect (Wan et al., 2013) selects edges to be sparsified randomly. (Ashouri et al., 2018) showed that the network performance can be maintained by eliminating insignificant weights without modifying the network architecture.

## 1.2 OUR CONTRIBUTIONS: OUTPUT INFORMED EDGE SPARSIFICATION

Following these works, we argue network sparsification can be a very effective tool for reducing sizes and complexities of DNNs and CNNs, without any significant loss in accuracy. However, we also argue that edge weights cannot be used "as is" for pruning the network. Instead, one needs to consider the *significance* of each edge within the context of their place in the network (Figure 2): "*Two edges in a network with the same edge weight may have different degrees of contributions to the final network output*" and in this paper, we show that it is possible to *quantify significance of each edge in the network*, relative to their contributions to the final network output and use these measures significance to minimize the redundancy in the network by sparsifying the weights that contributes insignificantly to network. We, therefore, propose a novel *iSparse framework*, and experimentally show, that we can sparsify the network, by almost 50%, without impacting the network performance. The *key contributions* of our proposed work are as follows:

- **Output-informed quantification of the significance of network parameters**: Informed by the final layer network output, iSparse computes and propagates edge significant scores that measure the importance of each edge with respect to the model output (Section 3).
- **Retraining-free network sparsification** (*Sparsify-with*): The proposed iSparse framework is robust to edge sparsification and can maintain network performance without having to retraining the network. This implies that one can apply iSparse on pre-trained networks, on-the-fly, to achieve the desired level of sparsification (Section 3.3)
- **Sparsification during training** (*Train-with*): iSparse can also be used as a regularizer during the model training allowing for learning of sparse networks from scratch (Section 4).

As the sample results in Figure 1b shows, iSparse is able to achieve 30-50% sparsification with minimal impact on model accuracy. More detailed experimental comparisons (See Section 5) of iSparse against PFEC, NISP, Retraining-Free and DropConnect on benchmark datasets illustrated that iSparse leads to more effective network sparsifications.

## 2 RELATED WORKS

A *neural network* is a sequence of layers of neurons to help learn (and remember) complex non-linear patterns in a given dataset (Grossberg, 1988). Recently, *deep neural networks* (DNNs), and particularly CNNs, which leverage recent hardware advances to increase the number of layers in the

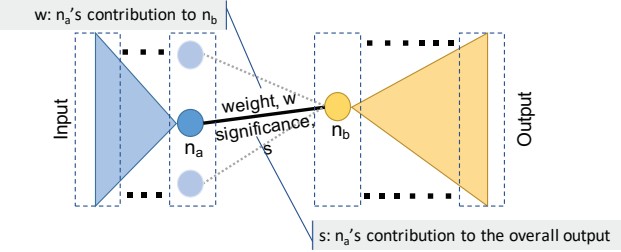

Figure 2: Overview of iSparse sparsification, considering the $n_i$'s contribution to overall output rather than only between $n_i$ and $n_j$ neurons

network to scales that were not practical until recently, (Lawrence et al., 1997; Yang et al., 2015; Hinton et al., 2012; Liang & Hu, 2015; Karpathy & et. at., 2014) have shown impressive success in several data analysis and machine learning applications.

A typical CNN consists of (1) *feature extraction layers* are responsible for learning complex patterns in the data and remember them through layer weights. An $m \times n$ layer, $\mathcal{L}_l$, maps an $n$-dimensional input ($Y_{l-1}$) to an $m$-dimensional output ($Y_l$), by training for a weight matrix $W_l \epsilon \mathbb{R}^{m_l \times n_l}$ (see Section 3.1 for further details); (2) *activation layers*, which help capture non-linear patterns in the data through activation functions ($\sigma$) which maps the output from a feature extraction layer to a non-linear space ( *ReLU* and *softmax* are commonly used activation functions); and (3) *pooling layers*, (sampling) which up- or down-sample the intermediate data in the network.

The training process of a neural network often comprises of two key stages: (1) *forward-propagation* (upstream) maps the input data, $X$, to an output variable, $\hat{Y}$. At each layer, we have $\hat{Y}_l = \mathcal{L}(Y_l) = \sigma(W_l Y_l + B_l)$, where $Y_1 = X$. Intuitively, each layer learns to extract new features from the features extracted by the prior layer. (2) *backward-propagation* (downstream) revises network weights, $W$, based on the training error, $Err = |Y - \hat{Y}|$, where $Y$ and $\hat{Y}$ are the ground truth and predictions of the model, respectively, such that update weights are defined as $W' = W - Err$.

The number of trainable parameters in a deep network can range from as low as tens of thousands (LeCun et al., 1999) to hundreds of millions (Simonyan & Zisserman, 2015) (Table 1 in Section 5). The three order increase in the number trainable parameter may lead to parameters being redundant or may have negligible contribution to the overall network output. This redundancy and insignificance of the network parameters has led to advancements in network regularization, by introducing dynamic or informed sparsification in the network.

These advancements can be broadly classified into two main categories: parameter pruning and parameter regularization. In particular, pruning focuses on compressing the network by eliminating the redundant or insignificant parameters. (Han et al., 2015; Han & et. al., 2016) aims to prune the parameters with near-zero weights inspired from $l_1$ and $l_2$ regularization (Tibshirani, 1996; Tikhonov, 1963). (Li et al., 2016) choose to filter out convolutional kernel with minimum weight values in given layer. Recently, (Yu et al., 2018) minimizes the change in final network performance by eliminating the neuron that have minimal impact on the network output by leveraging neuron importance score ($N_L$) (See Section 5.3) - computed using *Inf-FS* (Roffo & et. al., 2015). More complex approaches have been proposed to tackle the problem of redundancy in the network through weight quantization. (Rastegari & et. al., 2016) propose to the quantize the inputs and output activations of the layers in a CNN by using step function and also leveraging the binary operation by using the binary weights opposed to the real-values weights. (Chen & et. al., 2015) focuses on low-level mobile hardware with limited computational power, and proposed to leverage the inherent redundancy in the network for using hashing functions to compress the weights in the network.

(Bahdanau et al., 2014; Woo et al., 2018) showed that the each input feature to a given layer in the network rarely have the same importance, therefore, learning there individual importance (attention) helps improve the performance of the network. More recently, (Garg & Candan, 2019) has shown that *input data* informed deep networks can provide high-performance network configurations. In this paper, we rely on *output information* for identifying and eliminating insignificant parameters from the network, without having to update the edge weights or retraining the network.

## 3 iSparse: Output Informed Sparsification of Neural Networks

As discussed in Section 1, in order to tackle complex inputs, deep neural networks have gone increasingly deeper and wider. This design strategy, however, often results in large numbers of insignificant edges[1] (weights), if not redundant. In this section, we describe the proposed *iSparse*, framework which quantifies the significance of each individual connection in the network with respect to the overall network output to determine the set of edges that can be sparsified to alleviate network redundancy and eliminate insignificant edges. iSparse aims to determine the significance of the edges in the network to make informed sparsification of the network.

### 3.1 Mask Matrix

A typical neural network, $\mathcal{N}$, can be viewed as a sequential arrangement of convolutional ($\mathcal{C}$) and fully connected ($\mathcal{F}$) layers: $\mathcal{N}(\boldsymbol{X}) = \mathcal{L}_L \left( \mathcal{L}_{L-1} \ldots \left( \mathcal{L}_2 \left( \mathcal{L}_1(\boldsymbol{X}) \right) \right) \right)$, here, $\boldsymbol{X}$ is the input, $L$ is the total number of layers in the network and $\mathcal{L} \in \{\mathcal{C}, \mathcal{F}\}$, s.t., any given layer, $\mathcal{L}_l \mid 1 \leq l \in L$, can be generalized as

$$\mathcal{L}_l(\boldsymbol{Y}_l) = \sigma_l(\boldsymbol{W}_l \boldsymbol{Y}_l + \boldsymbol{B}_l) = \hat{\boldsymbol{Y}}_l, \tag{1}$$

where, $\boldsymbol{Y}_l$ is the input to the layer (s.t. $\boldsymbol{Y}_l = \hat{\boldsymbol{Y}}_{l-1}$ and for $l = 1$, $\boldsymbol{Y}_1 = \boldsymbol{X}$) and $\sigma_l$, $\boldsymbol{W}_l$, and $\boldsymbol{B}_l$ are the activation function, weight, and bias respectively. Note that, if the $l^{th}$ layer has $m_l$ neurons and the $(l-1)^{th}$ layer has $n_l$ neurons, then $\hat{\boldsymbol{Y}}_l \in \mathbb{R}^{m_l \times 1}$, $\boldsymbol{Y}_l \in \mathbb{R}^{n_l \times 1}$, $\boldsymbol{W}_l \in \mathbb{R}^{m_l \times n_l}$ and $\boldsymbol{B} \in \mathbb{R}^{m_l \times 1}$.

Given this formulation, the problem of identifying insignificant edges can be formulated as the problem of generating a sequence of binary **mask matrices**, $M_1, \ldots, M_L$, that collectively represents whether any given edge in the network is sparsified (0) or not (1):

$$\boldsymbol{M}_l = \mathbb{B}^{n_l \times m_l}, \mathbb{B} \in \{0, 1\}, 1 \leq l \leq L \tag{2}$$

### 3.2 Edge Significance Score

Let $M_l$ be a mask matrix as defined in Equation 2, and $M_l$ can be expanded as

$$\boldsymbol{M}_l = \begin{bmatrix} \mathbf{M}_{l,1,1} & \ldots & \mathbf{M}_{l,1,n_l} \\ \vdots & \ddots & \vdots \\ \mathbf{M}_{l,m_l,1} & \ldots & \mathbf{M}_{l,m_l,n_l} \end{bmatrix}, \tag{3}$$

where each $\mathbf{M}_{l,i,j} \in \{0, 1\}$ corresponds to an edge $e_{l,i,j}$ in the network. Our goal in the paper is to develop an edge significant score measure to help set the binary value of $\mathbf{M}_{l,i,j}$ for each edge in the network. More specifically, we aim to associate a non-negative real valued number, $\mathbf{E}_{l,i,j} \geq 0$, to each edge in the network, s.t.

$$\mathbf{M}_{l,i,j} = \begin{cases} 1 & \mathbf{E}_{l,i,j} \geq \tau_l(\theta_l) \\ 0 & \mathbf{E}_{l,i,j} < \tau_l(\theta_l) \end{cases}, \forall i = 1 \ldots m_l, j = 1 \ldots n_l. \tag{4}$$

Here, $\tau_l(\theta_l)$ represents the lowest significance of the $\theta_l\%$ of the most significant edges in the layer $l$. Intuitively, given a target sparsification rate, $\theta_l$, we rank all the edges based on their edge significance scores and keep only the highest scoring $\theta_l\%$ of the edges by setting their mask values to 1.

As we have seen in Figure 1a, the (signed) weight distribution of the edges in a layer is often centered around zero, with large numbers of edges having weights very close to 0. As we also argued in the Introduction, such edges can work counter-intuitively and add noise or non-informative information leading to reduction in the network performance. In fact, several existing works, such as (Ashouri et al., 2018), relies on these weights for eliminating insignificant edges without having to retrain the network architecture. However, as we also commented in the Introduction, we argue that edge weights should not be used alone for sparsifying the network. Instead, one needs to consider each edge within the context of their place in the network: Two edges in a network with the same edge weight may have different degrees of contributions to the final network output. Unlike existing works, iSparse takes this into account when selecting the edges to be sparsified (Figure 3).

---

[1]An edge is defined as a direct connection (weighted) between two neurons.

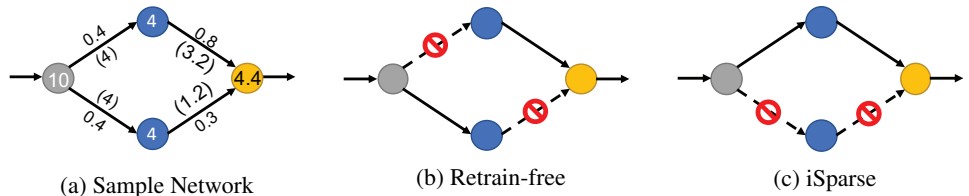

(a) Sample Network       (b) Retrain-free       (c) iSparse

Figure 3: A sample network architecture and its sparsification using Retraining-free (Ashouri et al., 2018) and iSparse; here node labels indicate input to the node; edge labels [0,1] indicate the edge weights; and edge labels between parentheses indicate edge contribution

More specifically, let $\boldsymbol{W}_l^+$ be the absolute positive of the weight matrix, $\boldsymbol{W}_l$, for edges in $l^{th}$ layer. We compute the corresponding edge significance score matrix, $\boldsymbol{E}_l$, as

$$\boldsymbol{E}_l = \boldsymbol{W}_l^+ \odot \boldsymbol{N}_l = \begin{bmatrix} \mathbf{W}_{l,1,1}^+ \times \mathbf{N}_{l,1} & \cdots & \mathbf{W}_{l,1,n_l}^+ \times \mathbf{N}_{l,n_l} \\ \vdots & \ddots & \vdots \\ \mathbf{W}_{l,m_l,1}^+ \times \mathbf{N}_{l,1} & \cdots & \mathbf{W}_{l,m_l,n_l}^+ \times \mathbf{N}_{l,n_l} \end{bmatrix}. \tag{5}$$

where, $\boldsymbol{N}_l$ represents the neuron significance scores[2], $\mathbf{N}_{l,1}$ through $\mathbf{N}_{l,n_l}$, and "$\odot$" represents the scalar multiplication between edge weights and neuron scores. $\mathbf{N}_{l,i}$, denotes the significance of the $i^{th}$ input neuron to the $l^{th}$ connection layer of the network, which itself is defined recursively, *based on the following layer in the network*, using the conventional dot product:

$$\boldsymbol{N}_l = \boldsymbol{W}_{l+1}^+ \boldsymbol{N}_{l+1} = \begin{bmatrix} \mathbf{W}_{l+1,1,1}^+ \times \mathbf{N}_{l+1,1} + \cdots + \mathbf{W}_{l+1,1,n_{l+1}}^+ \times \mathbf{N}_{l+1,n_{l+1}} \\ \vdots \\ \mathbf{W}_{l+1,m_{l+1},1}^+ \times \mathbf{N}_{l+1,1} + \cdots + \mathbf{W}_{l+1,m_{l+1},n_{l+1}}^+ \times \mathbf{N}_{l+1,n_{l+1}} \end{bmatrix}. \tag{6}$$

Note that $\boldsymbol{N}_l$ can be expanded as

$$\boldsymbol{N}_l = (\boldsymbol{W}_{l+1}^+(\boldsymbol{W}_{l+2}^+ \cdots (\boldsymbol{W}_{L-1}^+(\boldsymbol{W}_L^+ \boldsymbol{N}_L)))), \tag{7}$$

Above, $\boldsymbol{N}_L$ denotes the neuron scores of the final output layer, and $\boldsymbol{N}_L$ is defined using *infinite feature selection* (Roffo & et. al., 2015; Yu et al., 2018) as $\boldsymbol{N}_L = inffs(\hat{\boldsymbol{Y}}_L)$ where $\hat{\boldsymbol{Y}}_L \epsilon \mathbb{R}^{x \times n}$ ($x$ is the number of input samples and $n$ is the number of output neurons) to determine neuron importance score with respect to the the final network output. Given the above, the edge score (Equation 5) can be rewritten as

$$\boldsymbol{E}_l = \boldsymbol{W}_l^+ \odot \left(\boldsymbol{W}_{l+1}^+(\boldsymbol{W}_{l+2}^+ \cdots (\boldsymbol{W}_{L-1}^+(\boldsymbol{W}_L^+ \boldsymbol{N}_L)))\right). \tag{8}$$

Note that the significance scores of edges in layer $l$ considers not only the weights of the edges, but also the weights of all downstream edges between these edges and the final output layer.

### 3.3 EDGE SPARSIFICATION

As noted in Section 3.1, the binary values in the masking matrix $\boldsymbol{M}_l$ depends on $\tau_l(\theta_l)$, which represents the lowest significance of the $\theta_l\%$ of the most significant edges in the layer[3]: therefore, given a target sparsification rate, $\theta_l$, for layer $l$, we rank all the edges based on their edge significance scores and keep only the highest scoring $\theta_l\%$ of the edges by setting their mask values to 1. Note that, once an edge is sparsified, change in its contribution is not propagated back to the layers earlier in the network relative to the sparsified edge. Having determined the insignificant edges with respect to the final layer output, represented in form of the mask matrix, $\boldsymbol{M}_l$ (described in Section 3.1), the next step is to integrate this mask matrix in the layer itself. To achieve this, iSparse extends the layer $l$ (Equation 1) to account for the corresponding *mask matrix* ($\boldsymbol{M}_l$):

$$\mathcal{L}_l(\boldsymbol{Y}_l) = \sigma_l \left((\boldsymbol{W}_l * \boldsymbol{M}_l) \boldsymbol{Y}_l + \boldsymbol{B}_l\right), \tag{9}$$

---

[2] $\boldsymbol{N}_l$ summarizes the edge and neuron importance in the subsequent layers i.e. $\mathcal{L}_{l+1} \ldots \mathcal{L}_L$.

[3] In the experiments reported in Section 5, without loss of generality, we assume that $\theta_l$ has the same value for all connection layers in the network. This is not a fundamental assumption and iSparse can easily accommodate different rates of sparsification across connection layers.

Table 1: Number of trainable parameters (weights) and statistics for various benchmark datasets

| Network | LeNet | | | | | VGG | | | | | |
|---|---|---|---|---|---|---|---|---|---|---|---|
| Datasets | MNIST | FMNIST | COIL20 | COIL100 | NORB | CIFAR10 | CIFAR20 | CIFAR100 | SVHN | GTSRB | ImageNet |
| Weights | 44,426 | 44,426 | 62,556 | 69,356 | 168,801 | 32,418,834 | 32,428,844 | 32,508,924 | 32,418,834 | 38,743,323 | 138,357,544 |
| Statistics | | | | | | | | | | | |
| Resolution | $28 \times 28$ | | $32 \times 32$ | $32 \times 32$ | $96 \times 96$ | $32 \times 32$ | $32 \times 32$ | $32 \times 32$ | $32 \times 32$ | $32 \times 32$ | $224 \times 224$ |
| Train Set | 60,000 | 60,000 | 1300 | 6480 | 24,300 | 50,000 | 50,000 | 50,000 | ~73,000 | 39,209 | ~1,281,167 |
| Test Set | 10,000 | 10,000 | 140 | 720 | 24,300 | 10,000 | 10,000 | 10,000 | ~26,000 | 12,630 | ~50,000 |
| Labels | 10 | 14 | 20 | 100 | 6 | 10 | 20 | 100 | 10 | 43 | 1000 |

where, $*$ represents the element-wise multiplication between the matrices $\boldsymbol{W}_l$ and $\boldsymbol{M}_l$. Intuitively, $\boldsymbol{M}_l$ facilitates introduction of informed sparsity in the layer by eliminating edges that do not contribute significantly to the final output layer.

## 4 INTEGRATION OF iSPARSE WITHIN MODEL TRAINING

In the previous section, we discussed the computation of edge significance scores on a pre-trained network, such as of pre-trained ImageNet models, and the use of these scores for network sparsification. In this section, we highlight that iSparse can also be integrated directly within the the training process.

To achieve this, the edge significance score is computed for every trainable layer in the network using the strategy described in Section 3.2 and the mask matrix is updated using Equation 4. Furthermore, the back-propagation rule, described in Section 2, is updated to account for the mask matrices:

$$\boldsymbol{W}_l' = \boldsymbol{W}_l - \eta(\boldsymbol{M}_l * Err_l) \tag{10}$$

where, $\boldsymbol{W}_l'$ are the updated weights, $\boldsymbol{W}_l$ original weights, $\eta$ is the learning rate, and $Err_l$ is the error recorded by as the divergence in between ground truth ($Y_l$) and model predictions ($\hat{Y}_l$) as $Err_l = |Y_l - \hat{Y}_l|$. Note that, we argue that any edge that does not contribute towards the final model output, must not be included in the back-propagation. Therefore, we mask the error as $Err_l * \boldsymbol{M}_l$.

## 5 EXPERIMENTAL EVALUATION

In this section, we experimentally evaluate of the proposed iSparse framework using LeNet and VGG architectures (See Section 5.2) and compare it against the approaches, such as PFEC, NISP, and DropConnect (see Section 5.3).

### 5.1 SYSTEM CONFIGURATION

We implemented iSparse in Python environment (3.5.2) using Keras Deep Learning Library (2.2.4-tf) (Chollet et al., 2015)with TensorFlow Backend (1.14.0). All experiments were performed on an Intel Xeon E5-2670 2.3 GHz Quad-Core Processor with 32GB RAM equipped with Nvidia Tesla P100 GPU with 16 GiB GDDR5 RAM with CUDA-10.0 and cuDNN v7.6.4.[4].

### 5.2 BENCHMARK NETWORKS AND DATASETS

In this paper, without loss of generality, we leverage LeNet-5 (LeCun et al., 1999) and VGG-16 (Simonyan & Zisserman, 2015) as the baseline architectures to evaluate sparsification performance on different benchmark image classification datasets and for varying degrees of edge sparsification. In this section, we present an overview of these architectures (See Table 1).

**LeNet-5**: Designed for recognizing handwritten digits, LeNet-5 is simple network with 5 trainable (2 convolution and 3 dense) and 2 non-trainable layers using average pooling with *tanh* and *softmax* as the hidden and output activation. LeNet's simplicity has made it a common benchmark for datasets recorded in constrained environments, such as MNIST (LeCun et al., 1998), FMNIST (Xiao et al., 2017), COIL (Nene et al., 1996a;b), and NORB (LeCun et al., 2004).

---

[4]Results presented in this paper were obtained using NSF testbed: "*Chameleon: A Large-Scale Reconfigurable Experimental Environment for Cloud Research* (`https://www.chameleoncloud.org/`)"

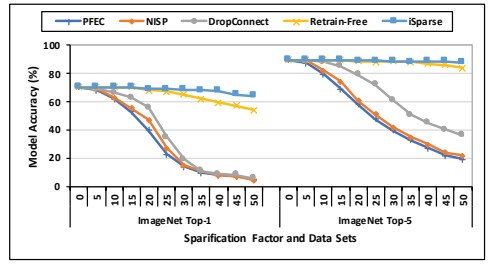
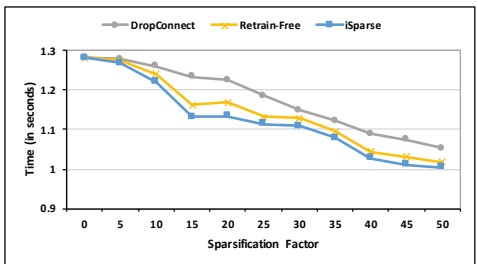

Figure 4: Top-1 and top-5 accuracy for spar- Figure 5: Model classification time vs sparsi-
sified VGG-16 for ImageNet (*sparsify-with*)  fication factor for MNIST dataset (*train-with*)

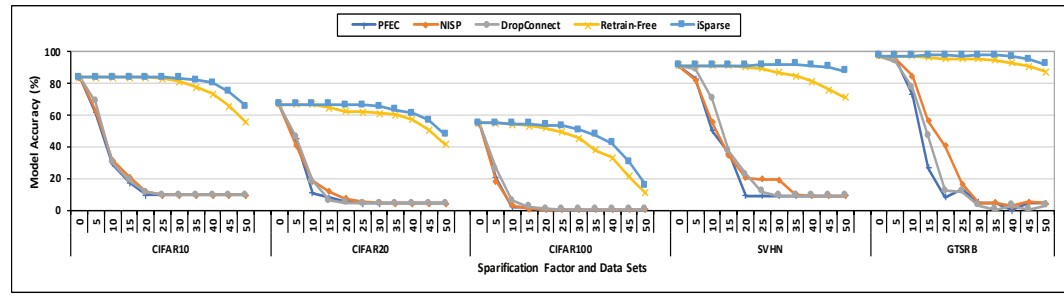

(a) VGG network (VGG-16)

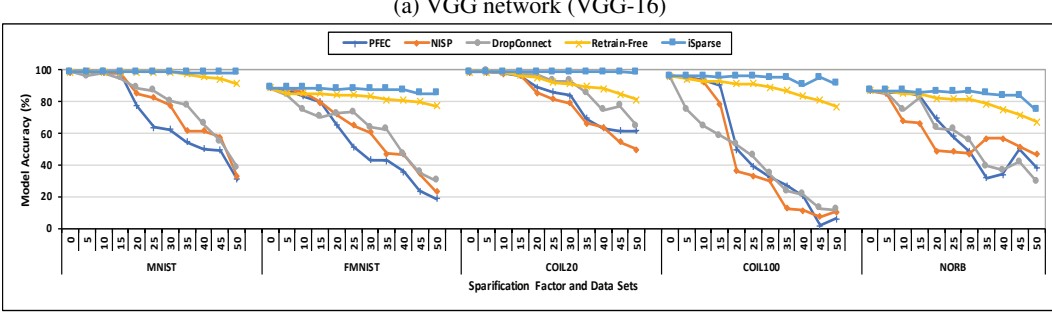

(b) LeNet network (LeNet-5)

Figure 6: Top-1 classification accuracy results for sparsified pre-trained models (*sparsify-with*)

**VGG-16**: VGG (Simonyan & Zisserman, 2015)'s, a 16 layer network with 13 convolution and 3
dense layers, with interleaved 5 max-pooling layers. VGG leverages ReLU as the hidden activation
to overcome the problem of vanishing gradient, as opposed to *tanh*. Given the ability of VGG
network to learn the complex pattern in the real-world dataset, we use the network on benchmark
datasets, such as CIFAR10/20/100 (Krizhevsky, 2009), SVHN (Netzer & et. al., 2011), GTSRB
(Stallkamp & et. al., 2012), and ImageNet (Deng et. al., 2009). Table 1 reports the number of
trainable parameters (or weights) for each model/data set pair considered in the experiments.

## 5.3 COMPETITORS

We compared iSparse against several state-of-the-art network sparsification techniques: **DropCon-
nect** (Wan et al., 2013) is a purely random approach, where edges are randomly selected for sparsifi-
cation. **Retraining-free** (Ashouri et al., 2018) considers each layer independently and sparsifies in-
significant weights in the layer, without accounting for the final network output contribution. **PFEC**
(Li et al., 2016) is a kernel pruning strategy that aims to eliminate neurons that have low impact
on the overall model accuracy. In order to determine the impact, PFEC computes the *l2*-norms of
the weights of the neurons and ranks them, separately, for each layer. **NISP** (Yu et al., 2018) pro-
poses a neuron importance score propagation (NISP) technique where neuron importance scores are
propagated from the output layer to the input layer in a back-propagation fashion.

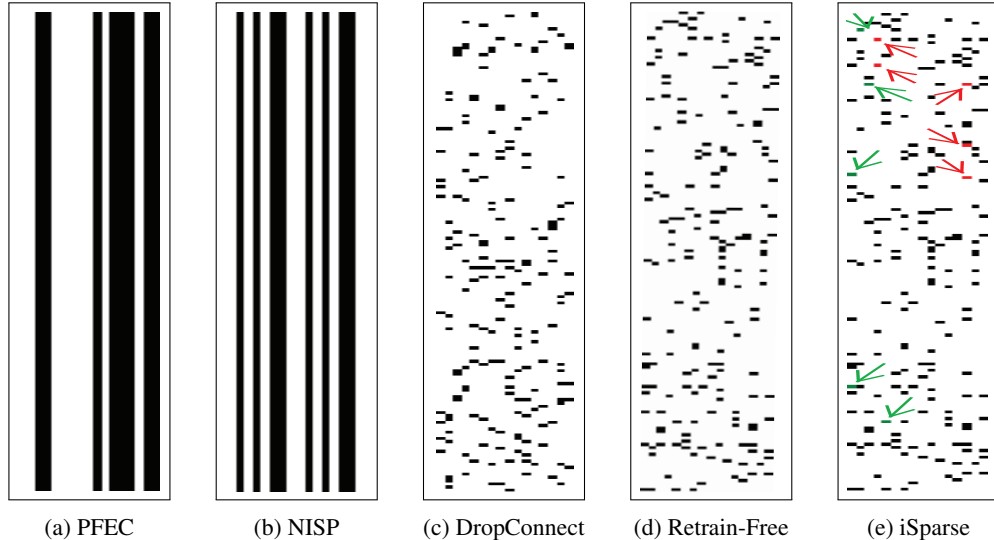

|            |            |                |                  |            |
|:----------:|:----------:|:--------------:|:----------------:|:----------:|
| (a) PFEC   | (b) NISP   | (c) DropConnect| (d) Retrain-Free | (e) iSparse |

Figure 7: Mask matrices for the LeNet network conv_2 layer for MNIST data (sparsification factor = 50%): dark regions indicate the edges that have been marked for sparsification; in (e) iSparse, the arrows point to those edges that are subject to different pruning decision from retrain-free in(d) (green arrows point to edges that are kept in iSparse instead of being pruned and red arrows point to edges that are sparsified in iSparse instead of being kept)

## 5.4 ACCURACY RESULTS

### 5.4.1 SPARSIFICATION OF PRE-TRAINED MODELS (*sparsify-with*)

In Figure 4, we first present top-1 and top-5 classification results for ImageNet dataset for VGG-16 network. As we see in the Figure 4, iSparse provides the highest robustness to the degree of sparsification in the network. In particular, with iSparse , the network can be sparsified by 50% with $\leq 6\%$ drop in accuracy for top-1 and $\leq 2\%$ drop in accuracy for top-5 classification, respectively. In contrast, the competitors, see larger drops in accuracy. The closest competitor, Retrain-free, suffers a loss in accuracy of $\sim 16\%$ and $\sim 6\%$ for top-1 and top-5 classification, respectively. The other competitors suffer significant accuracy drops after a mere 10-20% sparsification.

Figures 6a and 6b show the top-1 classification accuracy results for other models and data sets. As we see here, the above pattern holds for all configurations considered: iSparse provides the best robustness. It is interesting to note that DropConnect, NISP, and PFEC see especially drastic drops in accuracy for the VGG-16 network and especially on the CIFAR data. This is likely because, VGG-CIFAR is already relatively sparse (20% > sparsity as opposed to $\sim 7\%$ for VGG-ImageNet and $< 1\%$ for LeNet) and these three techniques are not able to introduce additional sparseness in a robust manner. In contrast, iSparse is able to introduce significant additional sparsification with minimal impact on accuracy.

Figure 7 provides the mask matrices created by the different algorithms to visual illustrate the key differences between the competitors. As we see in this figure, PFEC and NISP, both sparsify input neurons. Consequently, their effect is to mask out entire columns from the weight matrix and this prevents these algorithms to provide fine grain adaption during sparsification. DropConnect selects individual edges for sparsification, but only randomly and this prevents the algorithm to provide sufficiently high robustness. Retrain-free and iSparse both select edges in an fine-grained manner: retrain-free uses relies on edge-weights, whereas iSparse complements edge-weight with an edge significance measure that accounts for each edges contribution to the final output within the overall network. As we see in Figure 7 (d) and (e), this results in some differences in the corresponding mask matrices, and these differences are sufficient to provide significant boost in accuracy.

Table 2: Top-1 classification accuracy for sparsified different architectures and datasets (*train-with*)

| Datasets | MNIST | | | FMNIST | | | COIL20 | | | COIL100 | | | NORB | | |
|---|---|---|---|---|---|---|---|---|---|---|---|---|---|---|---|
| Factor | DC | RT | IS | DC | RT | iS | DC | RT | iS | DC | RT | iS | DC | RT | iS |
| Base (0%) | 98.79 | 98.79 | 98.79 | 88.02 | 88.02 | 88.02 | 95.52 | 95.56 | 95.56 | 90.00 | 90.00 | 90.00 | 84.42 | 84.42 | 84.42 |
| 5% | **98.90** | 97.99 | 98.09 | 88.23 | 88.70 | **88.75** | **95.00** | 94.16 | 94.72 | 89.83 | 90.12 | **90.61** | 84.51 | 84.75 | **85.03** |
| 10% | 98.17 | 98.45 | **98.56** | 87.60 | 87.02 | **87.89** | 94.72 | 95.01 | **95.55** | 90.05 | 89.81 | **90.33** | 85.35 | 85.98 | **86.25** |
| 15% | 98.76 | 98.79 | **98.92** | **88.14** | 87.43 | 87.96 | 95.00 | 95.21 | **95.50** | 89.50 | 89.73 | **90.00** | 83.91 | 84.12 | **84.73** |
| 20% | 98.79 | 98.81 | **98.98** | 88.02 | 88.00 | **88.11** | 95.83 | 95.95 | **96.10** | 90.83 | 88.54 | **91.03** | 83.04 | 82.56 | **83.29** |
| 25% | **98.85** | 98.85 | 98.80 | 87.62 | 87.54 | **88.19** | 95.27 | 94.66 | **95.80** | 90.11 | 89.98 | **90.22** | 85.33 | 85.01 | **85.61** |
| 30% | 98.78 | 98.71 | **98.98** | 88.00 | 87.75 | **88.25** | 95.00 | 94.74 | **95.00** | 90.12 | 90.12 | **90.55** | 83.69 | 85.91 | **86.08** |
| 35% | 98.73 | 98.73 | **98.79** | **88.14** | 86.99 | 88.06 | 94.72 | 94.99 | **95.27** | 90.11 | 91.35 | **91.99** | 83.98 | 83.01 | **84.81** |
| 40% | 98.85 | 98.97 | **98.99** | 87.81 | 87.25 | **88.30** | 95.00 | 95.31 | **95.56** | 88.22 | 90.01 | **90.44** | 83.14 | 84.12 | **85.08** |
| 45% | 98.91 | 98.95 | **98.99** | 87.45 | 87.11 | **87.82** | 95.29 | 95.04 | **95.56** | 89.45 | 89.87 | **90.23** | 85.47 | 85.12 | **86.02** |
| 50% | 98.55 | 98.85 | **99.00** | 87.76 | 88.15 | **88.46** | **95.00** | 94.16 | 94.72 | 90.66 | 90.21 | **90.85** | 84.49 | 85.61 | **86.77** |

| | VGG-16 | | | | | | | | | | | | | | |
|---|---|---|---|---|---|---|---|---|---|---|---|---|---|---|---|
| Datasets | CIFAR10 | | | CIFAR20 | | | CIFAR100 | | | SVHN | | | GTSRB | | |
| Factor | DC | RT | IS | DC | RT | iS | DC | RT | iS | DC | RT | iS | DC | RT | iS |
| Base (0%) | 84.14 | 84.14 | 84.14 | 66.96 | 66.96 | 66.96 | 55.09 | 55.09 | 55.09 | 91.33 | 91.33 | 91.33 | 97.68 | 97.68 | 97.68 |
| 5% | 84.21 | 84.56 | **84.66** | 62.26 | 62.01 | **62.85** | **57.35** | 52.74 | 53.10 | 93.49 | 92.42 | 92.70 | 96.49 | 97.68 | **98.64** |
| 10% | 84.35 | 84.33 | **84.35** | 63.63 | 66.53 | **67.33** | 51.32 | 51.01 | **51.48** | **95.00** | 92.56 | 92.96 | 95.03 | 91.09 | 93.24 |
| 15% | 83.43 | 83.51 | **83.69** | 60.95 | 59.87 | **62.49** | 52.18 | 52.53 | **52.97** | 91.96 | 93.11 | **93.40** | 96.83 | 95.91 | **97.29** |
| 20% | 82.83 | 85.04 | **85.81** | 61.79 | 63.45 | **65.03** | **51.67** | 50.01 | 50.52 | 92.63 | 93.34 | **93.74** | 93.02 | 94.41 | **95.76** |
| 25% | 84.07 | 84.32 | **84.56** | 61.85 | 60.19 | **62.52** | **56.47** | 53.25 | 53.81 | 82.56 | 87.14 | **87.63** | 95.11 | 96.66 | **97.76** |
| 30% | 83.41 | 84.17 | **84.48** | 62.14 | 58.39 | **62.14** | 51.19 | 54.05 | **54.50** | 91.95 | 91.97 | **92.26** | 97.18 | 97.43 | **98.56** |
| 35% | 83.06 | 84.95 | **85.42** | 60.09 | 57.81 | **61.79** | **52.05** | 54.42 | 49.03 | 92.58 | 92.38 | **92.65** | 97.18 | 97.61 | **98.17** |
| 40% | **83.96** | 82.01 | 82.21 | 66.34 | 65.09 | **66.62** | 50.19 | 55.11 | **55.68** | 88.79 | 91.92 | **92.31** | 95.61 | 96.91 | **97.61** |
| 45% | 82.75 | 82.81 | **82.83** | 65.65 | 66.93 | **68.72** | **54.18** | 53.47 | 53.63 | 91.38 | 91.23 | **91.62** | **96.71** | 93.31 | 94.02 |
| 50% | 83.44 | 84.15 | **84.53** | 65.37 | 65.04 | **68.82** | 51.65 | 51.78 | **52.56** | 82.94 | 87.61 | **87.68** | 94.31 | 95.91 | **97.31** |

Table 3: Robustness analysis for edge-based strategies vs iSparse (*train-with*)

| | Tan-Adam | | | Tanh-RMS | | | ReLU-Adam | | |
|---|---|---|---|---|---|---|---|---|---|
| | DC | RT | iS | DC | RT | iS | DC | RT | iS |
| MNIST | | | | | | | | | |
| Base (0%) | 98.82 | 98.82 | 98.82 | 98.79 | 98.79 | 98.79 | 98.89 | 98.89 | 98.89 |
| 5% | 98.90 | 98.23 | **98.94** | **98.90** | 97.99 | 98.09 | 98.66 | 98.91 | **98.95** |
| 10% | 99.02 | 98.75 | **99.15** | 98.17 | 98.45 | **98.56** | **98.90** | 98.75 | 98.86 |
| 15% | **98.94** | 98.01 | 98.83 | 98.76 | 98.79 | **98.92** | 98.94 | 99.01 | **99.23** |
| 20% | 98.76 | 98.41 | **98.97** | 98.79 | 98.81 | **98.86** | 99.05 | 99.32 | **99.53** |
| 25% | 98.91 | 98.25 | **98.92** | **98.85** | 98.78 | 98.80 | 98.89 | 98.85 | **98.96** |
| 30% | 99.01 | 98.04 | **98.92** | 98.78 | 98.71 | **98.78** | 98.88 | 98.74 | **98.94** |
| 35% | 98.94 | 98.65 | **99.05** | 98.73 | 98.75 | **98.79** | 98.86 | 98.62 | **98.92** |
| 40% | 99.12 | 98.88 | **99.57** | 98.85 | 98.97 | **98.99** | 98.83 | 98.23 | **98.95** |
| 45% | 98.84 | 98.42 | **99.04** | 98.91 | 98.95 | **98.99** | 99.07 | 99.01 | **99.48** |
| 50% | 99.07 | 98.01 | **99.12** | 98.55 | 98.85 | **99.00** | **98.98** | 98.16 | 98.89 |
| CIFAR10 | | | | | | | | | |
| Base (%) | 87.66 | 87.66 | 87.66 | 81.95 | 81.95 | 81.95 | 84.14 | 84.14 | 84.14 |
| 5% | 88.87 | 89.56 | **89.66** | 81.17 | 85.94 | **86.14** | 84.21 | 84.56 | **84.66** |
| 10% | 84.75 | 88.21 | **88.31** | 86.75 | 87.01 | **87.42** | 84.35 | 84.33 | **84.35** |
| 15% | 81.78 | 81.89 | **82.49** | 88.75 | 88.99 | **89.04** | 83.43 | 83.51 | **83.69** |
| 20% | 85.64 | 85.95 | **86.21** | 87.10 | 89.10 | **89.23** | 82.83 | 85.04 | **85.81** |
| 25% | 86.10 | 87.68 | **87.86** | 84.81 | 87.31 | **87.74** | 84.07 | 84.32 | **84.56** |
| 30% | **87.07** | 86.21 | 86.30 | 80.20 | 83.45 | **83.56** | 83.41 | 84.17 | **84.48** |
| 35% | 82.38 | 86.43 | **86.48** | 85.91 | 86.75 | **86.93** | 83.06 | 84.95 | **85.42** |
| 40% | 86.76 | 88.63 | **88.77** | 82.17 | 84.10 | **87.44** | **83.96** | 82.01 | 82.21 |
| 45% | 83.90 | 85.52 | **85.61** | 85.79 | 86.40 | **89.42** | 82.75 | 82.81 | **82.83** |
| 50% | **80.45** | 79.21 | 79.59 | 83.89 | 86.42 | **86.52** | 83.44 | 84.15 | **84.53** |

### 5.4.2 SPARSIFICATION DURING TRAINING (*train-with*)

Tables 2 present accuracy results for the scenarios where iSparse (iS) is used to sparsify the model during the training process. The table also considers DropConnect (DC) and Retrain-Free (RF), as alternatives. As we see in the table, for both network architectures, under most sparsification rates, the output informed sparsification approach underlying iSparse leads to networks with the highest classification accuracies.

### 5.5 ROBUSTNESS TO THE VARIATIONS IN NETWORK ELEMENTS

In this section, we study the effect of the variations in network elements. In particular, we compare the performance of iSparse (iS) against DropConnect (DC) and Retraining-Free (RF) for different

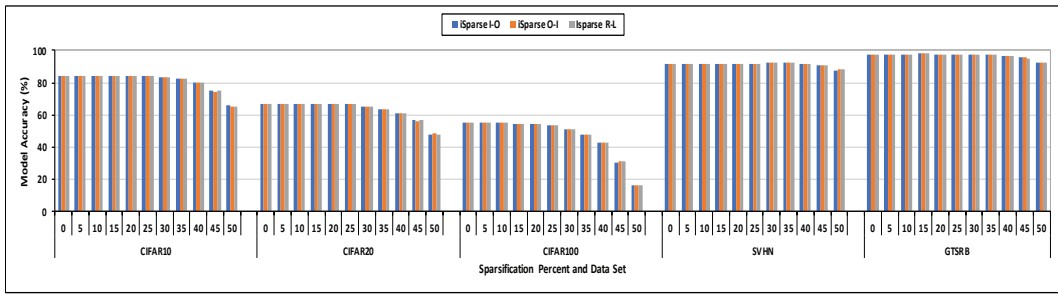

Figure 8: Robustness to the layer order while sparsifying the network with iSparse (*train-with*)

hidden activation functions and network optimizers. Table 3 presents classification performances for networks that rely on different activation functions (*tanh* and ReLU) and for optimizers (Adam and RMSProp). As we see in these two tables, iSparse remains the alternative which provides the best classification accuracy under different activation/optimization configurations.

### 5.6 ROBUSTNESS TO THE VARIATIONS IN SPARSIFICATION ORDER

We next investigate the performance of iSparseunder different orders in which the network layers are sparsified. In particular, we consider*three* sparsification orders: (a) *input-to-output layer order*: this is the most intuitive approach as it does not require edge significance scores to be revised based on sparsified edges in layers closer to the input; (b) *output-to-input layer-order*: in this case, edges in layers closer to the network output are sparsified first – but, this implies that edge significance scores are updated in the earlier layers in the network to account for changes in the overall edge contributions to the network; (c) *random layer order*: in this case, to order of the layers to be sparsified is selected randomly. Figure 8 presents the sparsification results for different orders, data sets, and sparsification rates. As we see in the figure, the performance of iSparse is not sensitive to the sparsification order of the network layers.

### 5.7 IMPACT OF SPARSIFICATION ON CLASSIFICATION TIME

In Figure 5, we investigate the impact of edge sparsification on the classification time. As we see in this Figure, edge sparsification rate has a direct impact on the classification time of the resulting model. When we consider that iSparse allows for $\sim 30 - 50\%$ edge sparsification without any major impact on classification accuracies, this indicates that iSparse has the potential to provide significant performance gains. What is especially interesting to note in Figure 5 is that, while all three sparsification methods, iSparse, DropConnect, and Retraining-Free, all have the same number of sparsified edges for a given sparsification factor, the proposed iSparse approach leads to the least execution times among the three alternatives. We argue that this is because the output informed sparsification provided by iSparse allows for more efficient computations in the sparsified space [5].

## 6 CONCLUSIONS

In this paper, we proposed iSparse, a novel output-informed, framework for edge sparsification in deep neural networks (DNNs). In particular, we propose a novel edge significance score that quantifies the significance of each edge in the network relative to its contribution to the final network output. iSparse leverages this edge significance score to minimize the redundancy in the network by sparsifying those edges that contribute least to the final network output. Experiments, with 11 benchmark datasets and using two well-know network architectures have shown that the proposed iSparse framework enables $30 - 50\%$ network sparsification with minimal impact on the model classification accuracy. Experiments have also shown that the iSparse is highly robust to variations in network elements (activation and model optimization functions) and that iSparse provides a much better accuracy/classification-time trade-off against competitors.

---

[5]Hardware configuration can be found in Section 5.1

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

Table 4: Model accuracies under different strategies for initializing neuron importance score - CI-FAR10 dataset - for different sparsification factors; Base = 0%

| Factor | VGG | | | ResNet | | |
| --- | --- | --- | --- | --- | --- | --- |
| | **IDN** | **PCA** | **InfFS** | **IDN** | **PCA** | **InfFS** |
| Base (0%) | 84.14 | 84.14 | 84.14 | 73.61 | 73.61 | 73.61 |
| 5% | 82.94 | 83.96 | **84.66** | 73.25 | 73.20 | **74.13** |
| 10% | 81.59 | 83.85 | **84.35** | 73.46 | 73.86 | **74.28** |
| 15% | 81.24 | 83.45 | **83.69** | 73.32 | 73.57 | **74.12** |
| 20% | 82.67 | 84.65 | **85.81** | 74.52 | 74.21 | **75.01** |
| 25% | 81.42 | 83.98 | **84.56** | 74.86 | 74.45 | **74.95** |
| 30% | 80.64 | 82.96 | **84.48** | 73.72 | 73.80 | **74.36** |
| 35% | 80.45 | 81.56 | **85.42** | 73.53 | 73.99 | **74.01** |
| 40% | 78.48 | 79.54 | **82.21** | 74.94 | 75.09 | **75.27** |
| 45% | 80.23 | 81.24 | **82.83** | 73.06 | 72.39 | **73.39** |
| 50% | 80.85 | 82.68 | **84.53** | 74.19 | 74.19 | **74.21** |

## A  APPENDIX

### A.1  ABLATION STUDIES FOR NEURON SCORE INITIALIZATION

In this section, we evaluate the performance on iSparse framework with different type of neuron score initialization strategies). We evaluate the following scoring strategies:

- Identity score (IDN) (all scores are "1")
- Principal Component Analysis (PCA) (Pearson (1901))
- Infinite Feature Selection (InfFS) (Roffo & et. al. (2015))

In Table 4, we observed that the iSparse framework with InfFS as a mechanism to determine the neuron scores for the final layers leads to a better performance opposed to the other strategies. Intuitively, this is because InfFS effectively identifies the importance of an individual features (neurons), while considering all possible subset of features. We believe this is a very highly desirable property: conventional feature selection strategies, such as PCA, ranks features based on their discriminatory power irrespective of how features relate with each other. Similarly, using same importance score (identity) treats each feature in a mutually exclusive fashion. In contrast, the way InfFS approaches the "feature selection" problem is highly appropriate to NNs, where the neuron output for the true label will be high, but we will also see low/small output for false labels.

### A.2  PERFORMANCE EVALUATION FOR THE RESNET ARCHITECTURE

In this section, we evaluate iSparse for the ResNet-18 architecture (He et al. (2016)) for both, sparsify-with (no retraining after sparsification) and train-with (retraining after sparsification) configurations Results are presented in Tables 5 and 6.

#### A.2.1  SPARSIFY-WITH FOR RESNET

As observed in Table 5, iSparse for ResNet demonstrates significant robustness to edge sparsification. iSparse outperforms DropConnect, and Retrain-Free, thus highlighting the importance of accounting edge's significance to subsequent network along with edge weight. .

#### A.2.2  TRAIN-WITH FOR RESNET

As we can see in the Table 6, the iSparse framework outperforms the competitors, such as Dropout, L1, DropConnect, Retraining-Free, and Lottery Winning Hypothesis (Frankle & Carbin (2019)). iSparse ability to account for edge significance based on both, edge weights and edge contributions to the final network output, leads to more robust and superior results.

Table 5: Model accuracy vs sparsification factor (*sparsify-with*) for ResNet architecture - CIFAR10

| Factor | DropConnect | Retraining-Free | iSparse |
|---|---|---|---|
| Base (0%) | 73.61 | 73.61 | **73.61** |
| 5% | 69.19 | 73.61 | **73.61** |
| 10% | 63.48 | 73.61 | **73.61** |
| 15% | 52.89 | 73.52 | **73.58** |
| 20% | 41.38 | 72.85 | **73.47** |
| 25% | 32.60 | 71.45 | **72.98** |
| 30% | 21.23 | 71.07 | **72.77** |
| 35% | 16.38 | 70.53 | **72.36** |
| 40% | 12.88 | 68.41 | **72.05** |
| 45% | 10.45 | 67.98 | **71.85** |
| 50% | 10.00 | 65.48 | **71.01** |

Table 6: Model accuracy vs sparsification factor (*train-with*) for ResNet architecture - CIFAR10

| Factor | Dropout | L1 | DropConnect | Retraning-Free | Lottery | iSparse |
|---|---|---|---|---|---|---|
| Base (0%) | 73.61 | 73.98 | 73.61 | 73.61 | 73.61 | 73.61 |
| 5% | 73.83 | N.A. | 73.72 | 74.35 | 74.10 | **74.43** |
| 10% | 74.52 | N.A. | 73.22 | 74.49 | 74.31 | **74.78** |
| 15% | 73.18 | N.A. | 73.69 | 73.82 | 74.40 | **74.12** |
| 20% | 75.42 | N.A. | 69.84 | 73.08 | 73.38 | **75.01** |
| 25% | 75.97 | N.A. | 72.43 | 73.75 | 73.77 | **74.95** |
| 30% | 74.69 | N.A. | 72.43 | 74.10 | 73.39 | **74.36** |
| 35% | 73.5 | N.A. | 72.59 | **74.07** | 73.99 | 74.01 |
| 40% | 67.89 | N.A. | 70.38 | 73.15 | 74.10 | **75.27** |
| 45% | 71.52 | N.A. | 67.07 | 72.37 | **74.15** | 73.39 |
| 50% | 66.2 | N.A. | 71.35 | 72.69 | 74.13 | **74.21** |

### A.2.3 ADDITIONAL TRAIN-WITH RESULTS FOR VGG

In Table 7, we present model accuracies for iSparse along with Dropout, L1, Retraining-Free, and Lottery winning hypothesis sparsification strategies. iSparse's informed sparsification leads to a superior performance in this model architecture as well.

Table 7: Model accuracy vs sparsification factor (*train-with*) for VGG architecture - CIFAR10

| Factor | Dropout | L1 | DropConnect | Retraining-Free | Lottery | iSparse |
|---|---|---|---|---|---|---|
| Base (0%) | 84.14 | 84.51 | 84.14 | 84.14 | 84.14 | 84.14 |
| 5% | 85.13 | N.A. | 84.21 | 84.56 | 84.29 | **84.66** |
| 10% | 84.77 | N.A. | 84.35 | 84.33 | 84.23 | **84.35** |
| 15% | 86.81 | N.A. | 83.43 | 83.51 | **84.88** | 83.69 |
| 20% | 77.81 | N.A. | 82.83 | 85.04 | 84.68 | **85.81** |
| 25% | 73.42 | N.A. | 84.07 | 84.32 | **84.87** | 84.56 |
| 30% | 76.16 | N.A. | 83.41 | 84.17 | 83.48 | **84.48** |
| 35% | 79.28 | N.A. | 83.06 | 84.95 | 84.79 | **85.42** |
| 40% | 77.51 | N.A. | 83.96 | 82.01 | 81.37 | **82.21** |
| 45% | 73.51 | N.A. | 83.96 | 82.01 | 81.37 | **82.21** |
| 50% | 69.21 | N.A. | 83.44 | 84.15 | 81.59 | **84.53** |

