# OpenReview forum: "iSparse: Output Informed Sparsification of Neural Networks"
_ICLR.cc/2020/Conference — Reject_

### Official Review · AnonReviewer2 · 2019-10-23
**Official Blind Review #2**

**Rating:** 1

**Review:**

The paper proposed a method for network sparsification based on the significance of each edge (weight). Unlike some of the existing works, edge significance in this work is explicitly defined based on their influence over the network output.

The algorithm can be summarized as follows:
1. Compute the significance of each activation (neuron) using an existing method (infinite feature selection).
2. Compute the significance of each edge as the product between the neuron significance and the absolute edge weight.
3. Sort the edges according to their significance scores and keep the top portion.

I'm mainly concerned about the limited technical novelty: the proposed technique is essentially a heuristic. Specifically, it is not clear why the significance of each edge should be defined according to equation (5) rather than some other forms. The intuition behind masking out the gradients of in-significant edges in (10), i.e., "we argue that any edge that does not contribute towards the final model output, must not be included in the back-propagation", is again a heuristic that lacks justification. If theoretical analysis is not possible, it might be necessary for the authors to conduct controlled experiments/ablation studies to show that some of the design choices made in the paper are indeed superior over other alternatives.

Another of my concern is that, if the goal is to make the sparsification decision aware of the network output (e.g., the value of the loss function), a simpler approach would be to enforce L1 regularization over the edges. This way, edges that do not lead to significant impact on the loss would be automatically pruned away. I wonder how would the proposed approach compare against this simple baseline.

Other suggestions:
* Empirical evaluation is conducted using LeNet and VGG16. It would be interesting to extend the analysis to some other seminal architectures, such as ResNet, Inception and MobileNets.
* It would be informative to report the hardware configuration used to obtain the execution time in Figure 5. Note the relative inference cost of different models may differ substantially over different hardware platforms.
* Writing of the paper can probably be polished further for better clarity.

**Experience Assessment:**

I have read many papers in this area.

**Review Assessment: Checking Correctness Of Derivations And Theory:**

I carefully checked the derivations and theory.

**Review Assessment: Checking Correctness Of Experiments:**

I assessed the sensibility of the experiments.

**Review Assessment: Thoroughness In Paper Reading:**

I read the paper at least twice and used my best judgement in assessing the paper.

---

> ### Author Response · Authors · 2019-11-14
> **Replies to the reviewer feedback**
>
> Q2.1- "It is not clear why the significance of each edge should be defined according to equation (5) rather than some other forms."
>
> A1.1-  We want to thank the reviewer for highlighting that this critical aspect of our contribution was not clear in the initial version of the manuscript. We observe that the conventional practice of sparsifying  edges (based on their weight values) does not properly account  their contribution the subsequent parts of the network. Therefore, we propose to learn the neuron importance with respect to the final network output explicitly and this is what Equation #5 formulates. As an empirical evidence of this, we can consider the "retraining-free" approach, which is essentially iSparse without neuron importance per Equation 5. Experiments reported in Figures 4 and 6, and Tables 3, 5, 6, and 7, show that  iSparse outperforms "retraining-free" for nearlly all sparsification factors in all configurations. We have revised the manuscript to better highlight these.
>
> Q2.2 - "if the goal is to make the sparsification decision aware of the network output (e.g., the value of the loss function), a simpler approach would be to enforce L1 regularization over the edges."
>
> A2.2-   While L1 regularization may be a simpler approach, it is not necessarily as effective as iSparse. The experimental results reported in Tables 6 and 7 in the revised manuscript demonstrate that the iSparse leads to more robust and informed regularization than L1. We evaluated L1 vs isparse for both VGG and ResNet architecture and, as the Tables 6 and 7, in the revised manuscript shows, iSparse outperforms L1 for most sparsification factors in VGG16 and almost all scenarios for ResNet.
>
> Q2.3- "Performance on ResNet architecture"
>
> A2.3-   We apologize for this oversight.  We have now included the evaluation of the ResNet architecture in the Appendix A of the revised manuscript. As Tables 5 and 6 show, iSparse outperforms the state of the art methods, such as, DropConnect, Retrain-Free, Lottery Winning Hypothesis, also for ResNet architecture.
>
> Q2.4 - "Hardware configuration for experimental evaluation"
>
> A2.4-   The hardware configuration was actually specified in the first paragraph of Section 5 in the original submission, but we apologize for not highligting this sufficiently; we now created a subsection (Section 5.1) specifically for the hardware configuration. To be precise, we implemented iSparse framework in Python environment (3.5.2) using Keras Deep Learning Library (2.2.4-tf) with TensorFlow Backend (1.14.0). All experiments were performed on an Intel Xeon E5-2670 2.3 GHz Quad-Core Processor with 32GB RAM equipped with Nvidia Tesla P100 GPU with 16 GiB GDDR5 RAM with CUDA-10.0 and cuDNN v7.6.4.

---

### Official Review · AnonReviewer3 · 2019-10-24
**Official Blind Review #3**

**Rating:** 3

**Review:**

This work proposes iSparse framework, which aims to sparsify a neural network by removing redundant edges. Unlike previous works where the edges were removed based on their weight value, or based on the relationship between input and output neurons, iSparse selects the edges to remove by computing the contribution of each edge with respect to the final outcome. The experiments show that, compared to several baseline methods, iSparse perform favorably on multiple datasets.

Comment:
Although I am not an expert in network pruning or network sparsification, I know that the Lottery Ticket Hypothesis (Frankle & Carbin, 2019) were able to remove at most 80% of the weights of neural networks (both fully-connected and ConvNets) and still retain the original performance level. Compared to that, iSparse's performance does not seem too impressive.

**Experience Assessment:**

I do not know much about this area.

**Review Assessment: Checking Correctness Of Derivations And Theory:**

I did not assess the derivations or theory.

**Review Assessment: Checking Correctness Of Experiments:**

I assessed the sensibility of the experiments.

**Review Assessment: Thoroughness In Paper Reading:**

I made a quick assessment of this paper.

---

> ### Author Response · Authors · 2019-11-14
> **Replies to the reviewer feedback**
>
> Q3.1 - ".. Lottery Ticket Hypothesis (Frankle & Carbin, 2019) were able to remove at most 80% of the weights of neural networks (both fully-connected and ConvNets) and still retain the original performance level. Compared to that, iSparse's performance does not seem too impressive."
>
> A3.1- We want to thank the reviewer for the pointer to  "Lottery Winning Hypothesis" work. However, as the following results (which are included in Tables 6 and 7 in the revised manuscript) show the difference between Lottery and iSparse is in fact substantial for most configurations, for both VGG and ResNet:
>
> Results for VGG
> Factor  | 5, 10, 15, 20, 25, 30, 35, 40, 45, 50
> Lottery | 84.29, 84.23, 84.88, 84.68, 84.87, 83.48, 84.79, 81.37, 81.93, 81.59
> iSparse | 84.66, 84.35, 83.69, 85.81, 84.56, 84.48, 85.42, 82.21, 82.83, 84.53
>
> Results for ResNet
> Factor  | 5, 10, 15, 20, 25, 30, 35, 40, 45, 50
> Lottery | 74.10, 74.31, 74.40, 73.38, 73.77, 73.39, 73.99, 74.40, 74.15, 74.13
> iSparse | 74.43, 74.78, 74.12, 75.01, 74.95, 74.36, 74.01, 75.27, 73.39, 74.21
>
> Moreover, iSparse can be applied to any trainable layer (dense or convolutional).

---

### Official Review · AnonReviewer1 · 2019-10-25
**Official Blind Review #1**

**Rating:** 3

**Review:**

This paper proposes a sparsification technique that seeks edges contributing negligible amounts to the performance of a network.

pros)
(+) This paper is written well but needs more clarity.

cons)
(-) This paper did not cite modern pruning/sparsification methods published recently.
(-) The proposed method has only compared with some outdated methods.
(-) Only LeNet-5 and VGG-16 have been used to validate the proposed method.
(-) This paper lacks any analysis of why the proposed methods would work well. Specifically, on the formulation of the sparsification method, it is hard to find sufficient backups why the authors did like that.

Comments)
- How would you guarantee using the infinite feature selection method could give proper score in general?
- How did you determine theta_l for each layer?
- The major problem of this paper is the experimental section. This paper only compared with outdated methods, so it is hardly verifying the effectiveness of the proposed method compared to other methods.
- It is necessary to involve ResNets as one of the baselines.

**Experience Assessment:**

I have read many papers in this area.

**Review Assessment: Checking Correctness Of Derivations And Theory:**

I assessed the sensibility of the derivations and theory.

**Review Assessment: Checking Correctness Of Experiments:**

I assessed the sensibility of the experiments.

**Review Assessment: Thoroughness In Paper Reading:**

I read the paper at least twice and used my best judgement in assessing the paper.

---

> ### Author Response · Authors · 2019-11-14
> **Replies to the reviewer feedback**
>
> Q1.1- “How would you guarantee using the infinite feature selection method could give proper score in general?”
>
> A1.1 - We would like to thank the reviewer for bringing to our attention that this was not clear in the paper. Below, we address this question both theoretically and empirically:
>
> + Theoretically, we pose the “score assignment” problem as a “feature selection” problem. The key intuition behind the choice of using infinite feature selection (InfFS) method is that InfFS has been shown to be effective in identifying the importance of an individual feature, while considering all possible subset of features. We believe this is a very highly desirable property: conventional feature selection strategies, such as PCA, ranks features based on their discriminatory power irrespective of how features relate with each other.  Similarly, using same importance score (identity) treats each feature in a mutually exclusive fashion. In contrast, the way InfFS approaches the "feature selection" problem is highly appropriate to NNs, where the neuron output for the true label will be high, but we will also see low/small output for false labels.
>
> + Empirically, in Table 4, we present an ablation study that compared InfFS against PCA and identity based methods, where we assess the performance on iSparse against different neuron importance scores initialisation strategies for N_L (final network layer). The results show clearly that the proposed framework (iSparse with InfFS) outperforms the other strategies.
>
> Q1.2- "How did you determine theta_l for each layer?"
>
> A1.2- We assume $theta_l$ is a user given parameter and its selection is outside of the scope of this submission. However, as we see in Tables 3, 5, 6, and 7, we observe that for almost all $theta_l$, iSparse outperforms the competitors, both in sparsify-with (no retraining after sparsification) and train-with (retrain after sparsification) setting.
>
> Q1.3- "The major problem of this paper is the experimental section. This paper only compared with outdated methods, so it is hardly verifying the effectiveness of the proposed method compared to other methods."
>
> A1.3-  In the original paper, we chose to compare our proposed iSparse framework with well-known neuron and edge (weight) sparsification strategies. We particularly focus on the edges sparsification strategies, DropConnect (ICML 2013), and Retraining-Free (NIPS 2018 Workshop CDNNRIA) as our proposed framework focus on edges not on neurons. However, taking into account this reviewer's valuable suggestion, we have now included comparisons against the  "Lottery Winning Hypothesis (LWH)" work, recently published at ICLR 2019. The results, presented in Tables 6 and 7, of the revised manuscript show that the proposed iSparse technique outperforms also this more recent approach in almost all scenarios.
>
> Q1.4- "It is necessary to involve ResNets as one of the baselines."
>
> A1.4- We apologize for this oversight  We have now included the evaluation of the ResNet architecture in the Appendix A of the revised manuscript. As Tables 5 and 6 show, iSparse outperforms the state of the art methods, such as, DropConnect, Retrain-Free, Lottery Winning Hypothesis, also for ResNet.

---

### Decision · Program_Chairs · 2019-12-19

**Decision:**

Reject

**Comment:**

Thank you very much for your feedback to the reviewers, which helped us a lot to better understand your paper.
However, the paper is still premature to be accepted to ICLR2020. We hope that the detailed reviewers' comments help you improve your paper for potential future submission.